# Effect of Pheromone-Mediated Mating Disruption on Pest Population Density of *Maruca vitrata* (Fabricius) (Crambidae: Lepidoptera)

**DOI:** 10.3390/insects11090558

**Published:** 2020-08-21

**Authors:** Onkarappa Dhanyakumar, Ramasamy Srinivasan, Muthugounder Mohan, Thiruvengadam Venkatesan, Kamanur Murali Mohan, Narayanappa Nagesha, Paola Sotelo-Cardona

**Affiliations:** 1University of Agricultural Sciences, GKVK Campus, Bengaluru, Karnataka 560065, India; dhanyakumarento@gmail.com (O.D.); entomurali@gmail.com (K.M.M.); nageshabt@gmail.com (N.N.); 2World Vegetable Center, 60 Yi-Min Liao, Shanhua, Tainan 7415, Taiwan; srini.ramasamy@worldveg.org; 3ICAR-National Bureau of Agricultural Insect Resources, Hebbal, Bengaluru, Karnataka 560024, India; mohan_iari@yahoo.com (M.M.); tvenkat12@gmail.com (T.V.)

**Keywords:** mating disruption, integrated pest management, pheromone blend, legume pod borer, mungbean

## Abstract

**Simple Summary:**

The legume pod borer is one of the most serious legume pests widely distributed in Asia, Africa, Australia, America, and Oceania. The use of synthetic pheromone lures has been developed as a more environmentally friendly alternative for its control. In this study, we evaluated the potential of the pheromone components as a mating disruption tool under laboratory and small-scale field conditions by identifying effective blends made out of single pheromone components or a different mix of them. The results from the laboratory experiment show that insects challenged with the blend ratio of 1:1:1 had lower fecundity and egg eclosion. A small-scale caged field experiment also showed a significantly disruption of normal mating with the above-mentioned ratio, leading to lower flower and pod damage, and higher mungbean yield. This study provides new evidence about the possibility to use pheromone components for mating disruption; however, more research is needed to determine appropriate ratios of pheromone blend to increase the effect of disruption. From an applied research perspective, more studies are needed to investigated the effectiveness of mating disruption strategy on different legume crops, dispenser types, release points in the field, and compatibility with conventional insecticides as part of an integrated pest management (IPM) combined approach.

**Abstract:**

The legume pod borer (*Maruca vitrata*) is one of the most serious legume pests due to its wide host range and high damage potential. Pheromone components on *M. vitrata* have been previously identified, allowing research on more environmentally friendly IPM tools for its control. *M. vitrata* produces a three-component pheromone blend containing (E, E)-10,12-hexadecadienal (major), (E, E)-10,12-hexadecadienol (minor), (E)-10-hexadecenal (minor). This study focused on the efficacy of synthetic pheromone lures and their blend components for mating disruption in *M. vitrata*. Under laboratory conditions, the mating behavior of *M. vitrata* pairs was observed from 18:00 to 02:00 h in an interval of 20 min to assess the efficacy of different pheromone lures. The scotophase behavior results show that the complete pheromone blend (E, E)-10,12-hexadecadienal + (E, E)-10,12-hexadecadienol + (E)-10-hexadecenal with a blend ratio of 1:1:1 effectively disrupted mating. The impact on mating disruption was evident from the lower fecundity and egg hatch/eclosion. The same lures were evaluated in a small-scale caged field study. The results show that the pheromone blend of (E, E)-10,12-hexadecadienal + (E, E)-10,12-hexadecadienol + (E)-10-hexadecenal in a1:1:1 ratio significantly disrupted the normal mating, leading to lower flower and pod damage and higher mung bean yield.

## 1. Introduction

Legume pod borer, *Maruca vitrata* (Fab.) (Lepidoptera: Crambidae) is widely distributed in Asia, Africa, Australia, America, and Oceania [1,2]. Indo-Malaysian is the most probable region of origin for the genus *Maruca*, including *M. vitrata*, which is found throughout the tropics [3]. *Maruca vitrata* is recognized as one of the most serious legume pests due to its wide host range, high damage potential and worldwide distribution [4].

*Maruca vitrata* larvae make webs on flower buds, flowers and pods, and once inside they start feeding on these plant parts [5]. Several legume crops including cowpea (*Vigna unguiculata* (L.) *Walp*.), mungbean (*Vigna radiata* (L.) Wilezek), black gram (*Vigna mungo* (L.) Hepper), lablab bean (*Lablab purpureus* (L.) Sweet), pigeon pea (*Cajanus cajan* (L.) Mills.), adzuki bean [*Phaseolus angularis* (Willd.) W.F. Wight (Syn. *Vigna angularis*)], common bean (*Phaseolus vulgaris* L.), the vegetable hummingbird tree (*Sesbania grandiflora* (L.) Pers.), the corkwood tree (*Sesbania cannabina* (Retz.) Pers.), and soybean (*Glycine max* (L.) Merr) are directly affected by the larval feeding of this species. Cowpea is one of its most preferred host plants in tropical Asia and Africa [6,7,8,9,10,11]. The severity of damage caused by *M. vitrata* led to yield losses recorded on cowpea, yardlong bean, and green gram in Thailand, west Sumatra, and Bangladesh [12,13,14]; 42–80% in India [15], Taiwan [16,17], and Brazil (soybean) [18,19]; and up to 100% yield losses in black gram in Karnataka, India [20].

Farmers rely almost exclusively on the application of chemical pesticides to combat *M. vitrata*. In 2010, farmers in Thailand and Vietnam applied approximately 16.3 kg/ha of pesticide per cropping cycle on yardlong bean [21]. Additionally, farmers in Cambodia use a mixture of four pesticides together in a single spray [22]. The challenge with *M. vitrata* is that the larvae are exposed on leaves only for a short time after hatching, and hence many farmers have to rely on periodical spraying throughout the growing season. Furthermore, more than 90% of the growers in Ratchaburi, Kanchanaburi, and Pathum Thani provinces in Thailand relied on chemical pesticides as a prophylactic measure for controlling pests on yard-long bean, and around 70% of growers applied pesticides once a week [23]. As a result, heavy dependence on conventional pesticides has led to many environment and health problems. However, pesticide use also failed to achieve a satisfactory level of control of *M. vitrata* [24] due to increasing pesticide resistance in this insect pest [25].

A more environmentally friendly alternative, synthetic sex pheromone lures, has been developed for monitoring, mass trapping and mating disruption of many insect pests of agricultural importance. In the particular case of *M. vitrata*, three pheromone components have been identified, with (E, E)-10,12-hexadecadienal (EE10,12-16:Ald) as the major sex pheromone component of *M. vitrata* [26]. In addition, (E, E)-10,12-hexadecadienol (EE10,12-16:OH); (E)-10-hexadecenal (E10-16:Ald) [27]; and (Z, Z, Z)-3,6,9-tricosatriene (ZZZ3,6,9-23:H) [28] have been recognized as the minor pheromone components. Interestingly, in in-field trapping experiments conducted in Benin, Ghana, India and Burkina Faso, the synthetic lures of EE10,12-16:Ald, EE10,12-16:OH and E10-16:Ald in the ratio of 100:5:5 (Benin ratio) was attractive to *M. vitrata* males, whereas the major compound alone was most effective in Burkina Faso [27,29]. However, when tested in Taiwan, Thailand, and Vietnam, the same components failed to attract the male moths [30,31]. More specifically, in India, in an experiment carried out under laboratory conditions, the pheromone blend (Z, E)-10,12-hexadecadienal + (E, E)-10,12-hexadecadienol + (Z)-10-hexadecenal (100:10:5) was found to elicit a better antennal response than the standard three-component blend [32].

*Maruca vitrata* is a genetically complex species, with the presence of multiple *Maruca* species or subspecies [4,8], and was also reported to have two forms in Australia [33]. There are three putative *Maruca* species, one in Latin America, one each in Oceania (including Indonesia) and in Asia, Africa, and two putative *M. vitrata* subspecies in Asia and Africa have also been reported [8]. A recent study found the presence of different *Maruca* species and/or subspecies in different continents based on the diversity within pheromone binding protein genes [34]. Since different species or subspecies appear to exist in the genus *Maruca*, the pheromone composition and reception of pheromone may not be uniform in different geographical locations. As previously mentioned, earlier studies on *M. vitrata* showed that under field conditions, the blend ratio of 100:5:5 was effective in Benin [29] as a monitoring tool but failed in the effective attraction of the adults in Southeast Asian countries. Therefore, our intention with this study is to gain a broader perspective on the appropriate ratio, therefore we tried with 1:1:1 ratio to understand what happens at the maximum expression of all components. Considering these facts, this study was conducted in order to investigate whether different pheromone blend ratios alone (1:1:1) or in combination affect the mating and reproduction of *M. vitrata*. First, the study assessed the efficacy of different pheromone alone/blend components in disrupting sexual communication, and their impact on fecundity and egg hatching of *M. vitrata* under laboratory conditions. Second, a small-scale field experiment was conducted to understand how different pheromone blends affect the fecundity of *M. vitrata*, host plant damage, and yield. From an applied perspective, the results from this research will allow us to have another environmentally sound pest management strategy to decrease the heavy reliance on chemical control.

## 2. Materials and Methods

### 2.1. Maruca vitrata Mass Culture under Laboratory Conditions

Second instar larvae of *M. vitrata* were obtained from a stock culture at the insect rearing facility of the World Vegetable Center, Shanhua, Taiwan. Insects were maintained in a 14 h light (L): 10 h dark (D) regime with a room temperature of 25 ± 2 °C and a relative humidity of 70 ± 10%. The larvae were kept in plastic containers (9 cm height × 9 cm width × 12 cm length), provided with an artificial diet [35], and reared until pre-pupation. In pre-pupal stage, corrugated paper was put inside the box as a pupation substrate. Later, pupae were transferred into mating cages (44 cm width × 45 cm length × 58 cm height) for 6–8 days, and the emerging adult moths were nourished with 10% glucose solution. During the oviposition period, *Sesbania grandiflora* (Fabaceae) leaves and flowers were provided as egg-laying substrate. The egg-laying substratewas kept in a plastic container, and provided with artificial diet [35]. As soon as the resulting neonates started feeding on the artificial diet [35], the remaining leaves and flowers were removed from the plastic container.

### 2.2. Efficacy of Pheromone Lures in Disrupting Sexual Communication of M. vitrata—Laboratory Studies

Scotophase behavioral assays were conducted to assess the efficacy of pheromone lures during the sexually active time of adult moths. For this purpose, late pupae (well-developed silken cocoon: N = 100) were separated and kept singly in cups measuring 2 cm × 1.5 cm, and provided with a drop of honey as nourishment for freshly emerged adults. The adults were sexed based on the abdominal characters, with a sharp or forked abdominal tip (male genitalia) feature used as the main distinguishing aspects of the male, while a female was recognized for having a blunt-tipped abdomen [28]. Later, two separate rooms with a temperature of 25 ± 2 °C and relative humidity of 70 ± 10% were used to directly compare a single lure treatment to a no pheromone control. The paired comparisons were conducted for each pheromone treatment separately over time using mating cages (44 cm width × 45 cm length × 58 cm height). Each treatment used 4 replicates (mating cages) in a dark room at one time. i.e., mating disruption lures as follows: T1 = (E, E)-10,12-hexadecadienal; T2 = (E, E)-10,12-hexadecadienol; T3 = (E)-10-hexadecenal; T4 = T1 + T2; T5 = T1 + T3; T6 = T2 + T3; T7 = T1 + T2 + T3 (alone/blend lures was tested @ 1 mg/lure. Source: Pest Control India Pvt. Ltd., Bengaluru, India), whereas in another room, the four mating cages were not exposed to any disruption lure (untreated control). The purity of each compound is as follows: (E, E)-10,12-hexadecadienal (98.3%), (E, E)-10,12-hexadecadienol (98.64%) and (E)-10-hexadecenal (99.02%). All the mating cages were provided with 10% glucose solution on cotton fibers as food source and *S. grandiflora* leaves or flowers as an egg-laying substrate. Four virgin male and female moths (<24 h old) were released into each of the four cages evaluated per treatment. Observation on mating was recorded continuously for eight hours at 20 min interval starting from 18:00 and until 02:00 h. To avoid light disturbance, observations were carried out with a LED 3W lamp with red cellophane during the scotophase up to 48 h after the release of adults (2 eight-hour sessions). The total number of mated pairs was recorded.

### 2.3. The Impact of Mating Disruption on Fecundity of Maruca vitrata

After 2 eight-hour sessions, the used male and female moths were removed from the cage. Later, the used female moths (N = 4/mating cage) were transferred from their respective individual mating cages (four replications per treatment) on to individual oviposition containers that were made from plastic cups measuring 15 cm × 5 cm. These separated females were nourished with 10% glucose solution in cotton fibers and provided with the *S. grandiflora* leaves, and flowers as egg-laying substrates. The egg-laying substrates were changed daily for the entire female longevity period. The number of eggs and neonate larvae was counted for each day separately.

### 2.4. Small Scale Caged Field Study

The experiment was conducted at World Vegetable Center farm in Shanhua, Taiwan (23°06′53.1″ N 120°17′53.5″ E). The experimental design was a randomized complete block design, with three blocks (field dimensions: 95 m × 32.5 m). Each block contained seven treatments plus an untreated control, as follows: T1 = (E, E)-10,12-hexadecadienal; T2 = (E, E)-10,12-hexadecadienol; T3 = (E)-10-hexadecenal; T4 = T1 + T2; T5 = T1 + T3; T6 = T2 + T3; T7 = T1 + T2 + T3 (alone/blend lures was tested @ 1 mg/lure. Source: Pest Control India Pvt. Ltd., Bengaluru, India). Furthermore, each treatment (i.e., pheromone lure) was located inside an insect net house (2.5 m width × 2.5 m length × 2.4 m height) and separated 10 m from other treatments [27]. Furthermore, a distance of 10 m × 10 m was maintained within and/or between treatments and replications. In addition, two rows of commercial corn were sown between treatments, with a spacing of 1 m × 30 cm. Within the selected field (2.5 m × 2.5 m), mung bean crop was sown and maintained. Seeds of mung bean (*Vigna radiata*) variety Tainan No-5 were sown with a spacing of 10 cm × 1 m. After thinning, 30 plants/treatment were maintained in each insect net house. Two rows of commercial corn were also sown as a border crop. The border crop and insect net houses were used to prevent the movement and oviposition of out crossed adult female *M. vitrata* moths into the experimental area. Ten pairs of <24 h old virgin M. vitrata adults were released at two times into the net-houses, first during the flower initiation stage (approx. 65 days after sowing), and second during the pod formation stage (approx. 75 days after sowing). One pheromone lure was installed at a height of 10 cm above the crop canopy level inside at the center of each net-house/small scale caged field. Pheromone lures were installed inside the net-houses one day before the release of *M. vitrata* adults, and one day ahead of the second release of adults (pod initiation stage). Observations on the percentage of flower damage (7–8 days after first release), percent pod damage (15 days after first release and 7 days after second release) and yield were recorded (15–20 days after second release).

### 2.5. Statistical Analysis

Mean and chi-square 2 × 2 contingency table tests of association were applied to compare the mated and un-mated adult pairs in the treatment and corresponding control cages during the laboratory experiment. The total number of eggs and larvae was analyzed in a paired t-test. In small scale field experiment, percent of flower, pod damage, and yield were analyzed with one factor ANOVA followed by Tukey’s test by using SAS (version 9.1; SAS Institute, Cary, NC, USA). Before statistical analysis, and to ensure data normality, the percentage of flower and pod damage was arc sin [*asin*(*sqrt*(*x*))] transformed, with *x* being the percentage of flower or pod damage.

## 3. Results

### 3.1. Efficacy of Pheromone Lures in Disrupting Sexual Communication of M. vitrata and Effect on Fecundity

Regarding disruption of sexual communication, the treatment with the three-component blend, T7 showed a significant mating disruption compared to their paired control treatment. Treatment T2 (i.e., minor component blend (E, E)-10,12-hexadecadienol) showed some disruption of mating but marginally failed to be statistically significant. The remaining blend treatments failed to cause mating disruption (Table 1).

In line with this, the effect on *M. vitrata* fecundity showed a reduction of 71% and 17% in the number of eggs laid by females in treatments T7 (i.e., three component blend) and T2 (i.e., minor component blend (E, E)-10,12-hexadecadienol), respectively and compared to their paired controls (Table 2). Furthermore, the egg hatching was also affected by the pheromone blends of these treatments, with a reduction of 85% and 22% in the larva recorded in treatments T7 and T2, respectively and compared to their paired control (Table 2). Interestingly, treatment T6 (i.e., T6 = T2 + T3) did not cause a reduction in eggs being laid by *M. vitrata* females, but instead a reduction of 25% in egg hatching was recorded (Table 2).

### 3.2. Small Scale Caged Field Study

Seventy percent less flower damage was observed in the three-component blend T7 compared to the control treatment. However, pod damage followed a different trend, with higher pod damage recorded in T1 (i.e., major pheromone component, (E, E)-10,12-hexadecadienal) compared to T4 (i.e., T1 + T2), which had the lowest pod damage across all treatments (Table 3). The three-blend component T7 had intermediate values, and did not differ from T1 and T4.

Significant differences were observed in the yield among the treatments (F_8,19_ = 6.79; *p* < 0.0024). More specifically, higher yield was recorded in the three-component blend (T7) compared to T2 (i.e., minor component blend (E, E)-10,12-hexadecadienol), T4 (i.e., major component T1 and + minor component T2), and T6 (minor component T2+ minor component T3), which were recorded as the treatments with the lowest yields. Furthermore, the yield in three-component blend T7 did not differ statistically from the untreated control treatment (Table 3). We must indicate here that flower and pod damage was recorded at the end of the experiment, and therefore damage assessments include the cumulative impact of two consecutive insect releases.

## 4. Discussion

Pheromone-mediated mating disruption is a vital tool in the suppression of insect pests [36]. Mating disruption is an eco-friendly practice and is safer for most non-target organisms including natural enemies, and is compatible with modern integrated pest management (IPM) [36,37]. The earlier studies on *M. vitrata* showed that the adult females produced three sex pheromone compounds: (E, E)-10, 12-hexadecadienal (major), (E, E)-10,12-hexadecadienol (minor) and (E)-10-hexadecenal (minor) [26,27], and these compounds showed higher attraction of moths (monitoring) under field condition with the blend ratio of 100:5:5 (called Benin ratio) in Benin [29] but failed in the effective attraction of the adults in Southeast Asian countries, such as Thailand and Vietnam, for the same components [31]. A previous study on two *M. vitrata* Chinese populations suggested sex polymorphism between these populations, since the sex pheromone from the Huazhou population was composed of (E)-10-hexadecenal, (E, E)-10, 12-hexadecadienal, and (E, E)-10,12-hexadecadienol in the ratio of 10.3:100:0.7, whereas the sex pheromone of the Wuhan population presented a different ratio of 79.5:100:12.1 [38]. In line with this finding, a recent study confirmed the presence of different *Maruca* species and/or subspecies in different continents [34] and differential production of sex pheromone compounds in different geographical locations [38,39], which is possible due to the presence of polymorphism in *M. vitrata* with respect to the pheromone production. With no doubt, the high pheromone polymorphism recorded in *M. vitrata* across diverse geographic areas in Africa, India, and Asia limits the potential use of the traps when used as monitoring or mass trapping tools.

Hence, in this study, we evaluated the potential of the pheromone components as a mating disruption tool under laboratory and small-scale field conditions by identifying effective blends made out of single pheromone components or a different mix of them. The previous studies on the monitoring of a *M. vitrata* population on different legume host crops in Taiwan failed to show the attraction for the same commercial pheromone component with the ratio of 100:5:5 [30]. In contrast, the current mating disruption study showed that the three-component blend (E, E)-10,12-hexadecadienal + (E, E)-10,12-hexadecadienol + (E)-10-hexadecenal (T7) in the ratio of 1:1:1 disrupted the mating significantly as compared to the untreated control. The total number of eggs, number of larvae, and percentage of eggs hatched provided information on the impact of mating disruption and the next generation population build-up. The three component blend (E, E)-10,12-hexadecadienal + (E, E)-10,12-hexadecadienol + (E)-10-hexadecenal (T7) and the individual minor compound (E, E)-10,12-hexadecadienol (T2) showed lower fecundity and larval hatchability due to lower or unsuccessful mating as compared to the untreated control. Similarly, *Lobesia botrana* (Tortricidae: Lepidoptera) females exposed to a synthetic species-specific pheromone (E7,Z9-12:Ac) showed lower fecundity than control females [40]. Among the mating disruption lures, the aldehyde component showed no impact on mating disruption with impact observed as an increase in eggs oviposited. In contrast, the *Amyelois transitella* (Pyralidae: Lepidoptera) synthetic species-specific pheromone containing aldehyde [(Z, Z)-11,13-hexadecadienal] alone compound is more effective for mating disruption [41]. In addition, the percentage of eggs that hatched indicates fertile and infertile eggs laid by the female moths. Hence, the overall percentage of eggs that hatched was lowest in T7 among the pheromone lures (Table 2).

Even though the adults were exposed to the major, and minor components alone and their combinations, mating was disrupted by the three-component blend of (E, E)-10,12-hexadecadienal + (E, E)-10,12-hexadecadienol + (E)-10-hexadecenal with the ratio of 1:1:1. The virgin females in the laboratory experiments/small scale field studies might also have released sex pheromones together with the synthetic pheromone lure, but the synthetic lure could have masked the pheromone released by females [42], since female sex pheromones are subjected to quality and quantity changes quite rapidly, which may be regarded as phenotypical condition-dependent traits [43]. Such changes in the released pheromone composition and/or its interaction with the synthetic lures could prolong the duration of mate finding and also disrupting the exact location of courtship partner. Similarly, *L. botrana* (being exposed to their synthetic species-specific sex pheromone (E7, Z9-12:Ac) reduced female calling, mating, and led to the suppression of next generation population density [40]. Hence, future studies should investigate the mechanisms of mating disruption, including the interaction between female-released pheromones and the synthetic lures. Though the three component blend (E, E)-10,12-hexadecadienal + (E, E)-10,12-hexadecadienol + (E)-10-hexadecenal (T7) and the individual minor compound (E, E)-10,12-hexadecadienol (T2) were capable of reducing the fecundity and larval hatchability of *M. vitrata*, the grain yield of mungbean did not differ significantly from the control. More fine tuning is probably required in the blends and dose in future research.

The current study provides new evidence about the possibility to use pheromone components for mating disruption. However, more basic research is needed to understand *M. vitrata*’s response in terms of intrasexual competition, specific mode of action (i.e., false trail following vs. confusion), competition between pheromone dispensers and natural pheromone, and sensory imbalance. More specifically, and as discussed and reviewed by [44], mating disruption needs to address areas such as intrasexual competition, the role of early male vs. late male responders and how they are attracted by “false females”, and an expected reduction in the ratio males to females. In this particular scenario, females would need to directly compete not only among them, but against the pheromone dispenser. Moreover, females would invest more in pheromone production, and unmated aging females in consequence would have a less reproductive potential compared to those young females able to mate under this pheromone competition scenario. In addition, more studies are needed to understand whether males respond to abundant false females/pheromone release points (i.e., false trails followers) or to a high dosage level. In any of these cases, mating disruption may decrease the odds of females being mated, but unless the efficacy of this disruption is high enough, even a low percentage of mated females will produce enough progeny to still cause substantial crop damage. The high dosage of pheromones provided during this experiment may also need to be further studied, since mating disruption may be caused by a sensory imbalance, interfering with the male’s ability to perceive and process the normal sensory inputs [45,46]. Low responsiveness in moths has been previously reported for the light-brown apple moth males of *Epiphyas postvittana* (Walker) (Tortricidae) to sex pheromones after pre-exposure [47,48]. Therefore, efforts are needed to investigate and understand the appropriate pheromone dosageto enhance the effectiveness of a mating-disruption strategy in the management of *M. vitrata*, and avoiding a reduced attraction/saturation to high rates of pheromone [49].

From an applied research perspective, more questions remain in terms of optimal ratio elucidation, as well as effectiveness on different legume crops, various dispenser types, dispenser density (release point) in the field, and compatibility with conventional insecticides as part of an IPM combined approach.

## 5. Conclusions

The current study provides new evidence about the possibility to use pheromone components for mating disruption. In line with this, and based on the results of these experiments, more research is needed to understand the appropriate ratios of the pheromone blend to increase the effect of disruption. Furthermore, the mechanisms of disruption should be examined using dosage response experiments with optimized pheromone blend. Such information will provide the optimal components necessary to successfully pursue mating disruption in the management of *M. vitrata* and less reliance on chemical insecticides. In addition, more studies are needed to optimize the use of this technology including dispenser types, density of release points, potential interference from host plants, release period, and compatibility with other IPM tools.

## Figures and Tables

**Table 1 insects-11-00558-t001:** (±SE) *Maruca vitrata* mating percentage and chi-square values (with Yates correction) under different mating disruption pheromone blends (treatments) under laboratory conditions (scotophase period) (N = 16 virgin pairs, <24 h old). T1= (E, E)-10,12-hexadecadienal (100%); T2 = (E, E)-10,12-hexadecadienol (100%); T3 = (E)-10-hexadecenal (100%); T4 = T1 + T2 (1:1); T5 = T1 + T3 (1:1); T6 = T2 + T3 (1:1); T7 = T1 + T2 + T3 (1:1:1). Mean mating values obtained from paired comparison experiment with four replicates, and four *M. vitrata* pairs per replicate.

Lure Treatment	Mean Mating %	Chi-Square	*p*-Value
T1	1.75 ± 0.25	1.143	0.2850
Control	2.75 ± 0.25		
T2	2.00 ± 0.41	3.636	0.0565
Control	3.50 ± 0.28		
T3	3.00 ± 0.41	0.948	0.3302
Control	3.75 ± 0.25		
T4	2.75 ± 0.25	0.731	0.3924
Control	3.50 ± 0.29		
T5	1.75 ± 0.25	1.143	0.2850
Control	2.75 ± 0.25		
T6	2.00 ± 0.00	0.518	0.4716
Control	2.75 ± 0.48		
T7	1.75 ± 0.25	7.127 *	***0.0076* * **
Control	3.75 ± 0.25		

* indicates the level of significance at 5%: degree of freedom = 1.

**Table 2 insects-11-00558-t002:** Mean (±SE) number of *Maruca vitrata* eggs, hatched larvae, and percentage of eggs that hatched from females exposed to seven different pheromone lures under laboratory conditions (scotophase period) (n = 16 females). T1 = (E, E)-10,12-hexadecadienal; T2 = (E, E)-10,12-hexadecadienol; T3 = (E)-10-hexadecenal; T4 = T1 + T2; T5 = T1 + T3; T6 = T2 + T3; T7 = T1 + T2 + T3. Total number of eggs, number of larvae, and hatchability from paired comparison experiment with four replicates, and four *M. vitrata* pairs per replicate.

Lure Component	Total Number of Eggs			Total Number of Larvae			% of Egg Hatchability
	Mean ± SE	*t*-Test	*p*	Mean ± SE	*t*-Test	*p*	
T1	778.25 ± 65.5	2.121	0.062	556.0 ± 59.8	1.249	0.151	71.0
Control	464.75 ± 88.9			396.0 ± 75.0			85.3
T2	1394.5 ± 155.9	3.876	***0.015* ***	1202.2 ± 130.0	4.403	***0.011* ***	86.3
Control	1688.7 ± 109.0			1546.5 ± 125.0			91.3
T3	1206.0 ± 314.4	−1.268	0.147	958.7 ± 289.5	−0.761	0.250	76.4
Control	838.0 ± 71.9			757.0 ± 71.9			90.1
T4	1338.0 ± 79.9	−0.409	0.355	958.7 ± 60.8	1.127	0.170	71.7
Control	1247.0 ± 175.0			1154.7 ± 171.1			92.3
T5	877.7 ± 154.6	−0.536	0.314	536.5 ± 119.0	0.687	0.270	58.7
Control	788.0 ± 38.2			638.5 ± 34.6			81.2
T6	1148.7 ± 80.8	0.445	0.343	753.2 ± 114.4	5.247	***0.006* ***	64.4
Control	1186.7 ± 160.7			1006.5 ± 91.2			86.6
T7	270.0 ± 12.9	4.921	***0.008* ***	120.7 ± 3.8	6.342	***0.004* ***	44.9
Control	840.5 ±114.1			802.7 ± 106.4			95.7

* indicates the level of significance at 5%.

**Table 3 insects-11-00558-t003:** Mean (±SE) flower and pod damage caused by *M. vitrata* and grain yield of mung bean crop. Means followed by different letters in a column are significantly different, as determined by Tukey’s post hoc test (*p* < 0.05).

Lure Treatment		Flower Damage (%)	Pod Damage (%)	Yield (g/2.5 × 2.5 m^2^) ^a^
T1	(E, E)-10,12-hexadecadienal	36.37 ± 6.98 ab	31.32 ± 4.62 a	59.93 ± 8.87 ab
T2	(E, E)-10,12-hexadecadienol	18.89 ± 2.30 ab	15.69 ± 1.82 ab	44.00 ± 6.00 b
T3	(E)-10-hexadecenal	29.46 ± 9.00 ab	20.19 ± 6.09 ab	--
T4	T1 + T2(1:1)	21.87 ± 4.20 ab	10.99 ± 2.41 b	43.83 ± 6.49 b
T5	T1 + T3(1:1)	14.53 ± 2.35 ab	14.35 ± 1.24 ab	61.60 ± 12.90 ab
T6	T2 + T3(1:1)	33.82 ± 3.31 ab	21.98 ± 4.41ab	39.30 ± 8.87 b
T7	T1 + T2 + T3(1:1:1)	13.16 ± 2.96 b	14.81 ± 1.11ab	91.47 ± 9.97 a
T8	Control	40.73 ± 5.1 a	21.52 ± 3.44ab	64.53 ± 7.38 ab

^a^ Only one block for T3 provided information on yield due to high precipitation during the harvesting period. Therefore, this treatment was not considered for the statistical analysis on the yield parameter.

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
