# Peer review of "Effect of Pheromone-Mediated Mating Disruption on Pest Population Density of Maruca vitrata (Fabricius) (Crambidae: Lepidoptera)"

_insects, 2020, doi:10.3390/insects11090558_

Round 1

Reviewer 1 Report

The paper “ Effect of Pheromone Mediated Mating Disruption on Pest Population Density of Maruca vitrata (Fabricius) (Crambidae: Lepidoptera)” provides a good introductory study on the basics of pheromone blend for disruption of this pest. The authors conducted two experiments to test determine what pheromone components might impact male moth behavior. Despite the importance of the findings, the study could have been improved with potentially more significant results by increasing replication in both studies.

Introduction

Much effort is given to explain the differences in response of Maruca vitrata from different geographic regions to a commercial pheromone.  The experiments reported relied on laboratory reared M.v. without reference to the apparent pheromone blend of the colony insects. It would also be good for the authors to provide rational why the ratio of the 3 component blend chosen was 1:1:1. The blend should also be reported early on in the methods  2.2 Scotophase study….(if it was indeed this blend used in the laboratory experiments)

2.2 Scotophase study

L121 – It is unclear if 4 virgin female moths were released into a single cage or if a single virgin female moth was released into each of 1 cages

L121 – Were the female moths added to cages containing virgin male moths?

L121 – replication?  I am assuming there is 1 virgin female moth in each cage and 4 cages were employed.  Please detail the experimental design.

Cage size/dimensions?

2.3 Impact of mating

2.3 same for experimental design. It is better to explicitly say. “The experimental design was a randomized complete block, or paired comparison,.. with N number of replicates”.

2.4 Semi-field experiment (Singular) this was 1 experiment with multiple treatments

Methods – The experiment consisted of a randomized complete block with 7 treatments replicated 3 times.

L144 – Ten pairs of VIRGIN M.v. adults were released…

Ten pairs is A LOT of moths in a small cage. Random interaction would be assumed with so many moths in close proximity. This is a high-pressure challenge to the MD formulations. You may have seen more separation in treatments in the results looking at Flower, pod, and yield assessments using half the pairs per cage which is still likely greater than the natural pest population. You may want to state this in the discussion section.

L149 – Pheromone lures 10m distance apart. More detail needed in methods. How many lures per cage? These were the same 1 mg lures as the lab experiment? Again, the field experiment is greatly challenging the effect of the pheromone with many moths (10 pairs applied twice, with I am assuming one 1mg lure). I would include this in the discussion. There may very well be greater differences either using less pairs of moths or a few more lures per cage. It is possible that this might especially be so regarding the flower and pod damage assessments. The current results are still valid, but T2 & T5 may also have an impact under natural conditions given the severity of the challenge presented herein.

Statistical analyses

Statistical analyses and the rules that define the tests are critical in any research. The authors conducted the bare minimum necessary to qualify for statistical analyses. In a search of pheromone experiments it is common to find laboratory experiments with treatments replicated 30+ times. This isn’t for mere redundancy, but to strengthen the analyses of measurements. Often replication in field experiments is much less than that of laboratory experiments, but still these this field experiment likely would benefit by increasing replication from 3 to 6. 

Conclusion

The conclusions are sparse and do not reflect the results of the experiments. The experiments give no indication to elucidating the mechanism of disruption. The authors are restating generalizations about pheromone mating disruption in general. The conclusions should be a direct reflection on the results of the manuscripts experimental results. This section needs much more attention to fleshing out what the results mean, the implication to M.v. disruption/control in the future, and potential what other experiments might be need next.

i.e. based on the results of these experiments more research is need to understand the appropriate ratios of pheromone blends to increase the effect of disruption. Furthermore, the mechanism of disruption should be examined using dosage response experiments using the optimized pheromone blend. Such information will provide the optimal components necessary to successfully pursue MD for M.V. management and less reliance on chemical insecticides….

The manuscript as written is unpublishable, but will be after the authors address the following. Please add complete detail to the experimental designs and address the low numbers of replication.  Provide a Conclusion. The current one is unacceptable a written.

Reviewer 2 Report

Effect of Pheromone Mediated Mating Disruption on 2 Pest Population Density of Maruca vitrata (Fabricius) 3 (Crambidae: Lepidoptera)

Dhanyakumar Onkarappa 1, Srinivasan Ramasamy 2, Muthugounder Mohan 3, Thiruvengadam 5 Venkatesan 3, Murali Mohan Kamanur 1, Nagesha Narayanappa 1 and Paola Sotelo-Cardona 2, *

The manuscript reports a study about whether different pheromone blend ratios alone or in combination affect the mating and reproduction of M. vitrata. They assessed the impact of the different pheromone components (alone or in combination) on fecundity and egg hatching of M. vitrata under laboratory conditions. In a second assay, a semi-field experiment assay, they study the effect of the same components on host plant damage and yield. It is a novel study since there are no previous studies about the possibility of mating disruption in M. vitrata.

The study demonstrates the affectation of the three pheromone components in the mating, the number of eggs and the percentage of their emergence. It also demonstrated the reduction of the percentage of damage with the consequent increase in harvest.

From my point of view, the manuscript provide new and useful information, so it deserves to be published after major revision of the text.

Page 2, line 48: there is an extra space

Page 2, line 60: I do not think “most importantly” was a correct expression. Pesticide resistance is not more important than health and environmental problems. Substitute this expression by “but also”

Page 2, line 64: there is an extra space.

Page 3, line 111: there is an extra space.

Page 3, line 117: was

Page 3, line 121: 4 “virgin moth into each cage” This means two virgin pairs in the cage? How many males and females in each cage? How many repetitions per treatment?  

Page 3, line 137. Specify that the treatments and lures are the same than in point 2.2.

Page 4. Line 114. “Ten pairs of M. vitrata adults were released…” change by “Ten pairs of <24 h old virgin M. vitrata adults were released”.

Page 4. Line 144. Delete “and released <24 h old virgin adults”

Page 4, line 146. Only one 1 mg pheromone lure was installed in the center of each white net house? Specify, please. Then the distance between the lures was 15 m? Or there were more than one lure per net house? One in each corner? And then the distance between the lures was 10 m as you said in Page 4, line 151.

Page 4, line 157.  There are two extra spaces.

Page 4, line 162. Add the study is in the scotophase.

Page 4, line 163: For T2 P>0.05. Is this difference significant?

Page 4, line 173: There is an extra space

Page 4, line 174: There is a very big difference among the mean number of eggs of the different control treatments. From 1688 to 464. It may be any explanation about? It seems if there are enough number of repetitions and the control conditions are constant, as in your case, the range of values should be narrower.

Page 5, line 178. Comment that all the treatments produce a reduction of the hatchability higher than their own control. Treatment T7 was the one with the higher reduction of hatchability. Discuss about in the discussion

Page 5, line 188. You do not comment the statistical differences found in flower and pod damage.

Page 6, line 246 to line 250. These can be the conclusions.

Page 7, line 252 to lines 258: These are not conclusions of your work. You did not studied the mechanism of mating disruption. This may be included in the discussion, Rewrite.  

Rewrite the conclusions and make the discussion more extensive.

Reviewer 3 Report

The authors have presented results from a laboratory and small-scale field study on the utility of 3 pheromone components either singly or in a 1:1 ratio blend as a mating disruption tool for M. vitrata. While there are some interesting results, the manuscript needs significant revision before being accepted for publication.  There are some main areas that need attention:

  1. Many sentences are not clear and suggested rewordings are provided below.
  2. The description of the laboratory experiments is not clear. Depending upon how this is addressed and how the experiment was actually conducted, the experiment itself may not be sound, e.g. it could suffer from cross-contamination of lures or lack of independence of the replicates
  3. Following from #2, the field experiment may also not be sound.  Having pheromone trials with only a 10 m spacing is likely not sufficient to prevent overlap of the treatments. The authors need to provide some evidence of the range of the influence of these components to justify this spacing. 
  4. authors are taking a P-value > 0.0500 as significant, which it is not. alpha < 0.05 is the criteria for determination of significance.  As such they are overstepping the reach of their experiments somewhat in their discussion and conclusions.

I hope the concerns expressed in items 2 and 3 are unfounded and that clarification is all that will be required.

Line 15 – reword…. ‘One eco-friendly IPM tool is pheromone-based mating disruption.’

Line 18 – reword…. ‘This study focussed on …’

Line 20 – indicate where these observations occurred, in the lab?

Line 23 – reorder…. ‘…. (E,E)-10,12-hexadecadienol effectively disrupted mating.’

Line 25 – move that these are laboratory results to line 23

Line 25 – ‘….semi-field net-house conditions as well.’

Lines 26-29 – reword ‘Results showed the pheromone blend of …. In a 1:1:1 ratio significantly disrupted normal mating, leading to lower flower and pod damage and higher mung bean yield.’

 Line 35 – start sentence with ‘Indo-Malaysia is the most probably region….’ And the sentence after ‘…throughout the tropics [3].’ Start new sentence with ‘Maruca vitrata is recognized as one…..’

Line 39 – reword…. ‘…flowers and pods, and feed on these plant parts.’

Line 40 – delete ‘Maruca vitrata causes extensive damage to’ and start sentence with ‘Several legume crops including <put list here> is caused by larval feeding of this species.’

Line 45 – don’t need ‘However’

Line 47-48 – sentence is repetitive.  Move the sentence on yield losses to earlier paragraph.  If you keep the 25-40% ‘wider adaptability’ you should define what this is.

Line 51 – replace ‘this pest’ with ‘M. vitrata’

Lines 52-53 – specify which year this occurred. ‘In <insert year here>, farmers in Thailand and Vietnam applied approximately 16.3 kg/ha of pesticide per cropping cycle on Yardlong bean. Additionally, farmers in ….’

Line 54 – start the sentence ‘Unfortunately, M. vitrata…’ with ‘The challenge with M. vitrata is that the larvae are exposed…’

Line 55-56 – is there no degree day model for this pest? When you say ‘periodical spraying’ do you mean ‘calendar spraying’? what is the range of egg hatch:(1 week? 3 weeks?

Lines 56-58…. Roll this sentence in with the previous sentence – they’re saying the same thing.

Line 59 – delete ‘a’… ‘As a result, heavy dependence upon conventional pesticides has led to many….’

Line 64 – replace ‘has’ with ‘have’

Line 71 – Southeast Asian

Line 72 – specify earlier in the sentence that the India work was done in the laboratory

Line 77 – you have ‘Oceania’ twice in the same sentence

Line 94 – ‘larvae were’  If they were housed singly, then state that… ‘The larvae were housed individually in plastic containers….’

Line 96 – ‘after the pre-pupal stage’?  doesn’t that mean they are pupae then? If the substrate needs to be added before they develop to pupae, then it might be ‘During the pre-pupal stage, a corrugated….’

Line 103 – I don’t think you need ‘Scotophase study to assess the’ in the heading.  Simply use ‘Efficacy of pheromone lures in disrupting sexual communication of M. vitrata – laboratory studies’.  More helpful to  know where these occurred with the details of when during the light cycle in the methods (which you did on Lines 122-124).

Line 105 – first line can be deleted – in an ‘Insects’ journal, we should know this

Line 107 – ‘active’ not ‘activated’

Line 108 – ‘…kept singly in cups measuring 2 cm x 1.5 cm, and…’

Line 109 – is it not possible to sex the pupae?  Were there any errors in sexing of the adults?

Line 111 – describe how the rooms were ‘isolated’?  by walls? By a distance of X metres? What measures were taken to ensure the least amount of contamination between treatments?

Lines 113-115 – were all these lures happening at the same time in the same room?  Not clear how the rooms were used.  I can easily read it as 2 rooms – 1 used for the ‘control’ (no lure) and the other room for the ‘treatment’ (one of the 4 lure options), but I can also read it as 1 room for the control and the other with all 4 lures happening at the same time.  Please clarify the set up, the number of replicates used for the treatments, and the time span when these experiments occurred, e.g. over a 6-week period? Over a 3-day period?.  Also clarify that the 48h really means 2 eight-hour observation sessions…. Correct?  And if so, please modify Line 127 to be the same.  ‘2 eight-hour sessions’ is different than a ’48 h of mating behavior observations’

Line 122 – not clear what the ‘paired treatments’ are or how these are set up – in this line it sounds like the paired treatments were in the SAME room, when I think I understand your treatments to be in 2 different rooms.  How many males were in each cage?

Line 124-125 – need to clarify your replicates…. ‘total number of mated pairs recorded’ in each session? Over both sessions?

Line 127 – remove ‘scotophase’

Line 129 – here it seems as though 1 female was in a mating cage while above it reads as 4 females/cage – which is it?

Line 133 – ‘…neonate larvae from each female were counted daily.’

Line 135 – what is a ‘white net house condition’?  need to describe this more clearly.  Was a white net (need details where this was sourced, size of the weave) draped over a block?  What was the crop?  Move this detail earlier in the paragraph.

Lines 137-139 – not sure 10 m is enough distance for a pheromone lure trial.  Do you have any literature/evidence to indicate that there would be no interference between the plots at this close distance.  For host volatile experiments – the distance is 20 m between traps and the range of influence is much less than that of a pheromone. 

Line 142 – was the outside border crop also corn? Should state this for clarity

Line 144 – here you state ’10 pairs of adults’, then later you state that there were < 24 h old virgin adults – are these the same adults?  Need to clarify this.

For this experiment – how are you creating the ‘mating disruption’? a single lure? Line 146 says ‘lures’ but you don’t specify how many in a net-house.  Section needs to be reworked for clarity.

Line 147 – ‘….at a height of 10 cm…’

Line 148 – ‘…and virgin adults (< 24 h old) released.’

Lines 151-153 – you made 2 releases into the same net-houses? The damage you are reporting is cumulative? Do you have any treatments where the moths were released only early or only late?  Is there a chance the presence of damage earlier in the season impacted the amount of damage later in the season?

Line 156 – here you state that there were ‘mated, un-mated adult pairs’ yet this is not made clear in the methods of the lab experiment.  Please ensure consistency between the methods, the data you collected and the analyses conducted on these data.

Line 156 – call it the ‘laboratory experiment’ rather than ‘scotophase period’

Line 157 – ‘….analysed using a paired t-test.’  Also ‘Percentage of flower, pod damage….’

Line 160 – for your transformation, state what your ‘x’ value was

Line 163-165 – a P value of 0.05 is NOT significant.  It has to be < 0.0500.  T2 is greater than this.

Table 1 caption – ‘Mean (+/- SE) number of Maruca vitrata mating…’   If you did a chi-square you should be presenting the NUMBER of mating pairs (as indicated in your methods).  How did you generate a percentage? If you totalled the number of matings across all treatments done on the same day (?) Across all replicates (?)  then please clarify how you generated these percentages and present the number of total matings in the caption so the reader can know what the percentages were based on.

Table 2 caption – reword needed… ‘Mean (+/- SE) number of M. vitrata eggs, hatched larvae and percentage of eggs that hatched from females exposed to 7 different pheromone lures. T1…..’

Line 183 – the studies were done in the field so I’d suggest renaming them ‘Small scale caged field studies’ and that will better describe what they are and where they were conducted.

Table 3 – indicate the ratio of the components in brackets, e.g. T1 + T2 (1:1)

Line 202 – delete the extra words ‘it is’, ‘of the’, ‘components’…. ‘… is an eco-friendly practice and safer to most non-target organisms including…’

Line 206 – delete ‘pheromonal compounds’ before the [26-27] reference and ‘pheromone’ before the next ‘compounds’

Line 210 – delete ‘had mainly’

Line 211 – replace ‘other IPM component/tool…’ with ‘focused on using these compounds for mating disruption.’

Line 215 – reword…. ‘Hence, in this study we evaluated the potential of the pheromone components as a mating disruption tool under laboratory and small scale field conditions by ….’

Line 221 – if the previous blend was a ratio of 100:5:5, why did you choose 1:1:1?  Should provide some background and rationale for this choice. Suggest to say that the blend ‘reduced number of mating pairs’ which is what you observed.

Line 223 – delete this last part of the sentence.  Your P-value doesn’t support the claim.

Line 224 – reword ‘egg hatch percentage’ with ‘percentage of eggs that hatched’

Line 231 – use ‘….the aldehyde component showed no impact on mating disruption with impact observed as an increase in eggs oviposited.’

Lines 232-234 – sentence is bulky and not clear

Line 235 – ‘indicates’ not ‘indicated’ and ‘eggs that hatched’, or more correctly ‘larvae that eclosed’

Line 238 – ‘the males were disturbed’…. Reword with…. ‘mating was disrupted by the 3-component…’

Line 240 – replace ‘mating cages’ with ‘laboratory experiments’ and ‘semi-field condition’ with ‘small scale field studies’

Lines 240-243 – bulky sentence, not clear what you are trying to say here

Line 243 – replace ‘in prolong’….with ‘to prolong’

Line 246 – ‘provides’ not ‘provided with’

Line 247 – reword ‘use the pheromone blend as a mating disruption strategy’ with ‘use pheromone components for mating disruption.’

Line 248 – replace ‘elucidate an optimal blend ratio’ with ‘elucidate the optimal ratio of the components’

Line 252 – is the ‘false-trail-following’ the mechanism observed for this species?

Line 258 – mating disruption does not, de facto, lead to a reduction in the number of larvae hatched.  It can result in fewer eggs laid.  Viability of the eggs is a separate issue.

Line 260 – do not use ‘etc.’, specify or just end the list.

Round 2

Reviewer 1 Report

The authors have made significant improvements to the manuscript. It reads much better especially with regard to the clarity of the experimental designs

Minor edits provided.

is L110 – The methods could still be improved for clarity, but the authors have made good progress in this test. 

Consider:

Later, two separate rooms with a temperature of 25 ± 2oC and relative humidity of 70 ± 10% were used to directly compare a single pheromone lure treatment to a no pheromone control. The paired comparisons were conducted for each pheromone treatment separately over time using mating cages (44 cm width × 45 cm length × 58 cm height).  Each treatment used 4 replicates (mating cages) in a dark room at one time.

L270 - Moreover, females would investment more in pheromone production

Reviewer 3 Report

The authors have done a reasonable job addressing many of the concerns raised by the reviewers.  There are some areas where the authors did not adequately address the comments from the reviewers and as such, further revision is still required.

Lines 74-85 – this was mentioned in the earlier review. Although the authors have indicated that previous component ratios were not effective, they failed to explain why they are using a 1:1:1 ratio, even though they provided this in their response to reviewers (but did not transfer this to the manuscript).

Line 85 – replace ‘semi-field experiment’ with ‘small scale field experiment’ to be consistent throughout the manuscript

Line 140 – distance between treatments. This item was raised in the first review and has not been adequately addressed by the authors.  In their response they provide the following:

“Thanks for your question, we have found a wide range of distances being consider in pheromone trap studies:

In case of Gypsy moth inter trap distance ranging from 2.5m to 40m (Elkinton and Cardé, Environ.Entomol.17(5):764-769 (1988) DOI: 10.1093/ee/17.5.764). In case of Coffee leaf minor Leucoptera coffeella spacing was maintained about 2.5m to 30m. (Bacca et al. Entomol. Exp. Appl.119: 39-45 (2006)  https://doi.org/10.1111/j.1570-7458.2006.00389.x). Grapholita molesta followed 10 m distance, (Arioli et al. Chilean J. Agric. Res. 74: 184-190 (2014) http://dx.doi.org/10.4067/S0718-58392014000200009)”

The authors could have used Bacca et al. to justify their distancing as that study found spacing closer than 10 m to show interference.  Conversely, Kong et al. (2014) for G. molesta found distances closer than 30 m to show interference, and Sharov et al. 2002 found gypsy moth to respond to display mating disruption behavior up to 250 m away from a block where this technique was employed.  The authors need to do a better job of justifying their selected 10 m distance as the 3 studies they have selected to validate their distance only shows this support in 1 case.

Line 152 – replace ‘one day earlier to’ with ‘one day ahead of the second release’.  The authors did not address the question initially asked about the damage and are making an assumption that any damage done earlier by the first release had no effect on damage observed following the second release. While it is difficult to know whether damage by the same species will impact oviposition and subsequent damage from the next release, the authors should at least acknowledge that this is possible and indicate that their damage assessments following the second release could be impacted by any damage present after the first release.

Line 159 – ‘In semi-field….’ Replace with ‘In the small scale field studies, percent of…’

Line 162 – state the value of ‘x’ used in the transformation

Line 251 – Sentence is not clear.  Are you saying that the presence of the synthetic lures may have affected the virgin females pheromone production? Or that the virgin females also producing pheromone resulted in the synthetic lure being less effective?

Round 3

Reviewer 3 Report

The authors have addressed the bulk of the comments in their latest version. I am particularly pleased with their justification of their use of the 10 m spacing. Inclusion of this reference in the beginning would have removed any doubt about their study design.  A few final details..... 

  1. Lines 80-84 - just some grammatical touch-ups.  'As previously mentioned, earlier studies on M. vitrata showed a blend ratio of 100:5:5 was effective under field conditions [29] as a monitoring tool but failed to attract adults in Southeast Asian countries. Our intention was to use a ratio of 1:1:1 to understand what happens when all components are equal in the blend. Our study investigated whether the pheromone components alone, or in combination in equal ratio would affect the mating behavior and reproductive performance of M. vitrata.'
  2. Line 162 - this one is my fault for not being clear. Typically when describing a transformation, the equation is presented as follows:  asin(sqrt(x+0.1)).  I was after the value they added to their variable prior to the square-root function.  It was not present in the original manuscript and their notation suggests that 'x' is the value - when based upon their response here 'x' is their variable (flower or pod damage).... but what was added to this 'x'?
  3. Line 251 - sentence is still not clear. The first part fo reference [49] is fine, but the connection to the second part of the second (and reference 42) does not make sense.  What is the impact on your study of the synthetic lure masking the 'real' pheromone produced by the live females? 
